# Apparent Molecular Weight Distributions in Bituminous Binders

**DOI:** 10.3390/ma15134700

**Published:** 2022-07-05

**Authors:** Giovanni Polacco, Miriam Cappello, Giacomo Cuciniello, Sara Filippi

**Affiliations:** Department of Civil and Industrial Engineering, University of Pisa, Largo Lucio Lazzarino, 1, 56122 Pisa, Italy; miriam.cappello@ing.unipi.it (M.C.); giacomo.cuciniello83@yahoo.it (G.C.); sara.filippi@unipi.it (S.F.)

**Keywords:** bitumen, apparent molecular weight distribution, gel permeation chromatography, linear viscoelastic functions, delta method

## Abstract

Molecular weight distributions are widely used to evaluate the effects of aging or modifiers in bituminous binders. As with polymers, the most common techniques to obtain the distributions can be subdivided into two main groups, depending on whether or not they use a solvent. In the first group, the dimension of the molecules is evaluated in a diluted unperturbed state, while, in the second, the dimension derives from the bulk, where aggregated or interacting molecules may behave as single entities. However, the calibration curves used in the bulk are tuned in order to homogenize the results derived from the two approaches. This sort of contradiction, plus the high number of experimental uncertainties, suggest that the term “apparent” should be used for both distributions. These aspects are well known in the field of polymers but have received less attention in the case of bitumens, which are even more complex. This paper pinpoints the advantages and disadvantages of the two techniques, thus highlighting the most appropriate use. Bulk methods are preferred when evaluating properties that are strictly dependent on the microstructure, such as the level of aging and the effects of additives or modifiers. Diluted methods should be used when the molecular size matters, such as in quantifying the presence of polymers or rejuvenators. Both techniques should be used for comparative studies only.

## 1. Introduction

The molecular weight distribution (MWD) influences the rheological and mechanical properties of viscoelastic materials, such as bitumen. For this reason, there is an increasing use of the MWD in the scientific literature related to asphalt paving, where it is correlated with the composition, performance, aging susceptibility and compatibility with additives and polymers [1]. The MWD can be determined by several experimental techniques, such as size-exclusion chromatography (SEC), field-flow fractionation and mass spectrometry [2]. Of these, the most popular in terms of bitumen is SEC, also known as gel permeation chromatography (GPC) [3,4]. In GPC, the data are collected in a diluted state, where the interactions among molecules are eliminated or at least strongly inhibited by the solvent. This therefore provides a distribution that corresponds to a disaggregate state and contributes little to the open debate on the complex structure of binders [5]. 

It would thus be helpful to obtain the distribution directly in bulk, as in the “inverse mechanical” approach, which uses rheological data, which are typically collected in the linear viscoelastic region. The idea is that the viscoelastic functions contain the complete spectrum of relaxation times, which can thus be deconvoluted into an MWD [6,7,8]. The procedure uses different viscoelastic functions and requires a correlation between the operating variable (i.e., reduced frequency, shear rate, time) and the molecular weight (MW).

One of the criteria adopted to derive such a correlation is to obtain a similar MWD from the GPC and inverse mechanical method. In other words, the calibration curves in the inverse mechanical approach are tuned in order to unify the results obtained with the two techniques [9,10]. This is somewhat questionable because molecules that tend to aggregate can be interpreted as a single entity in bulk and as multiple components in solution.

In summary, the two main approaches are very different from both the theoretical and experimental perspectives, and, consequently, in this paper, we differentiate between “solution” (SMWD) and “bulk” (BMWD) molecular weight distributions. In fact, neither approach guarantees a reliable quantitative description of the molecular weight distribution, and both should be considered as “apparent” molecular weight distributions (AMWDs) [11,12,13]. Because neither of the two distributions is a “real” one, the most appropriate use for both is as a comparison. This may be a comparison between two polymers with the same chemical composition, or between the distributions of the same material (polymer or bitumen) in different conditions, such as before and after aging. 

Despite the abovementioned uncertainties, the AMWD provides useful information and has been very effective in the study and characterization of bituminous binders. Based on this premise, this paper focuses on the following: (1) the adherence of the two distributions to the theoretical significance of the MWD; (2) the main differences between the two options; (3) the correlation between the two distributions; (4) guidance in selecting the most appropriate distribution.

## 2. Molecular Weight Distributions

The idea of the MWD derives from polymeric materials that are composed (at least in the case of homopolymers) of molecules with the same chemical composition, but variable lengths and structures (e.g., the branching degree). Simple knowledge of the chemical formula and average molecular weight is therefore insufficient to fully characterize the material and its properties. For example, in the case of polyolefins, the average MW may not correspond to a probable chain length because it is common practice to develop bimodal distributions with a low-MW fraction that reduces the viscosity in the melt state, and a high one that improves the mechanical properties of the product. Processing can also be performed on several types of machinery, each with specific requirements. This is why, and especially for the so-called “commodities”, there are many commercial grades of the same polymer with almost identical chemical compositions, but different MWDs.

There are several possible definitions for the MWD, and the interested reader can find an accurate description in any book on polymer science [2,14]. The two most commonly used are the *number molecular weight* distribution, which considers the molar fraction (*n(M)*) of molecules with molar mass (*M*), and the *weight molecular weight* distribution, which considers the weight fraction (*w(M)*). The two distributions clearly need to be normalized, and any average value can be easily derived from the moments of the distribution curves: (1)$$\int_{0}^{\infty }w\left(M\right)dM=1$$
(2)$${\bar{M}}_{n}=\frac{1}{\int_{0}^{\infty }w\left(M\right)\left(\frac{1}{M}\right)dM}$$
(3)$${\bar{M}}_{w}=\int_{0}^{\infty }w\left(M\right)MdM$$
where ${\bar{M}}_{n}$ and ${\bar{M}}_{w}$ are the number and weight average molecular weights, respectively.

Despite the continuous nature of Equations (1)–(3), the MWDs of a polymeric material are intrinsically discontinuous because the length of each molecule must be equal to an integer number of repeating units. It is thus common practice to use a discrete representation that takes into account fractions that correspond to molecules containing “*i*” repeating units, with *I* = 1, 2,…, ∞. If *M*_0_ is the molar mass of the repeating unit, then the two average molecular weights can be expressed as:(4)$${\bar{M}}_{n}=\overset{\infty }{\underset{i=1}{\sum }}n_{i}M_{i}=\frac{\sum_{i=1}^{\infty }N_{i}M_{i}}{\sum_{i=1}^{\infty }N_{i}}$$
(5)$${\bar{M}}_{w}=\overset{\infty }{\underset{i=1}{\sum }}w_{i}M_{i}=\frac{\sum_{i=1}^{\infty }N_{i}M_{i}^{2}}{\sum_{i=1}^{\infty }N_{i}M_{i}}$$
where *M_i_* = *iM*_0_ is the molar mass, and *N_i_* are the moles of molecules with *i* repeating units. 

In the case of bituminous materials, which are composed of molecules with different chemical compositions, the discontinuous nature of the distributions is different because there is no repeating unit that is common to all molecules. However, Equations (4) and (5) remain valid if *i* refers to a specific molar mass instead of the chain length. In addition, any experimental technique used to evaluate the distribution produces a finite set of data, which justifies the use of discrete expressions.

Other definitions of the average molar mass and distributions specifically derive from the technique used for their determination (e.g., the measure of the intrinsic viscosity in a given solvent, the separation by ultracentrifugation or the quantification of end groups). When referring to average values or distributions, it should therefore be specified which one is intended, as they can differ significantly from each other. For example, in the case of the step polymerization and stoichiometric feed of the reacting functional groups, the following theoretical expression can be easily derived from a statistical theory [14]:(6)$$n_{i}=\left(1-x\right)x^{i-1}$$
(7)$$w_{i}=i{\left(1-x\right)}^{2}x^{i-1}$$
where *x* is the conversion of the functional groups. Equations (6) and (7) represent the “most probable distribution”, or “Flory distribution”, for which the following average values can be determined:(8)$${\bar{M}}_{n}=\frac{{\bar{M}}_{0}}{\left(1-x\right)}\;{\bar{M}}_{w}=\frac{{\bar{M}}_{0}\left(1+x\right)}{\left(1-x\right)}\;DI=\frac{{\bar{M}}_{w}}{{\bar{M}}_{n}}=1+x$$

*DI* is the “dispersion index”, which, in this case, tends to the value of 2 for high conversions and has a direct correlation with the variance (σ) of the distribution:(9)$$\frac{{\bar{M}}_{w}}{{\bar{M}}_{n}}=\frac{\sigma^{2}}{{\bar{M}}_{n}{}^{2}}+1\geq 1$$

The higher the difference between the number and weight average molecular weights, the more dispersed the distribution. Of course, the dispersion index does not contain information on the shape and modes of the curves. Figure 1 refers to Equations (6) and (7) and shows how the number and weight distributions can differ from each other.

The number molecular weight distribution is a monotonically decreasing function, which thus indicates that, at any value of the conversion, the most probable length is 1, corresponding to an unreacted monomer. In contrast, the weight average distribution shows a maximum in the *i*-axis.

## 3. Main Differences between Solution and Bulk Distributions

### 3.1. Gel Permeation Chromatography

The main idea of GPC is to separate the molecules based on their hydrodynamic volume in solution. The apparatus consists of a column filled with a stationary phase, with a distribution of pore sizes. When the polymer solution (mobile phase) flows through the column, the paths of the molecules inside the stationary phase depend on their dimension. Large molecules are excluded from the small-size pores and follow a shorter path than small molecules. Therefore, the larger the molecule, the shorter its retention time, and vice versa. After the column, the solution passes through a detector, the signal of which gives the weight fraction of the solute as a function of the retention time. This is then converted into a weight molecular weight distribution by applying a calibration curve that correlates time and molar mass. 

The three main assumptions of the method are:“size exclusion”, which means that the stationary phase works as a molecular sieve, without chemical interactions with the mobile phase;The detector records all the molecules;A calibration curve is available.

These points have been widely discussed in the literature; however, it is useful to underline some of the differences between polymers and bitumen.

#### 3.1.1. Size Exclusion

Unfortunately, a complete absence of interactions between molecules and the stationary substrate is almost impossible, which may strongly influence the retention time. In the case of homopolymers, all the repeating units have the same chemical composition, and thus the magnitude of the interactions is in proportion to the molecular size. In contrast, in a bituminous binder, the molecules differ in both dimension and chemical composition, with the latter usually subdivided into a small number of families based on their polarity, such as the saturates, aromatics, resins and asphaltenes (SARA) fractions proposed by Corbett [15]. The different nature of the molecules affects both their hydrodynamic volume and their tendency to bond with the substrate. For example, paraffinic molecules have a higher hydrodynamic volume than aromatic ones of the same mass and thus elute sooner [11]. At the same time, the polarity determines the chemical interactions with the substrate, and highly interacting molecules remain for a longer time (or even forever) in the column. Both artefacts alter the supposed univocal correlation between the molecular weight and retention time.

#### 3.1.2. Detector

There are many types of detectors, but the most common ones in the bituminous literature are based on the refraction index (RI), ultraviolet absorption (UV) and light scattering (LS). While an appropriate choice is straightforward when dealing with polymers, the same is not true for bitumen. The wide spectrum of chemical compositions can induce errors because a detector can have different sensitivities or can even be completely blind to specific molecules. For example, consider the case of a UV detector, which has to be set on a single wavelength, and for which the value should be tailored to the specific solvent and solute.

#### 3.1.3. Calibration Curve

An appropriate number of monodispersed samples with known MWs is needed to obtain a calibration curve, which, for a given solvent, correlates the retention time and MW. In order to prevent problems related to the abovementioned uncertainties, the ideal situation is to have samples of the same polymer that needs to be tested. Of course, this is rarely possible, and reliable standards are only available for polystyrene (commonly used in the case of bitumen) and poly(ethylene oxide). For a limited number of polymers, the problem of the standards can be overcome by using “universal calibration”, which is based on Einstein’s viscosity formula and is exploited in combination with the Mark–Houwink equation [2]. The concept of universal calibration clearly cannot be applied to the case of bituminous materials, the compositions of which are uncertain and variable, depending on the source. For this reason, data are often reported as a function of the elution time, without risking the conversion to an MW.

A simple demonstration of the above considerations is given by the raw GPC spectra reported in Figure 2, which show the absorbance (in arbitrary units) of an unmodified base 50/70 Pen-grade bitumen as a function of the elution time, with both UV and IR detectors.

The two distributions show evident differences due to the sensitivity of the detectors with respect to the chemistry of the bitumen molecules. The small peak at the elution time of around 6 min in the RI spectrum corresponds to buthylhydroxytoluene (BHT), which is added as stabilizer in the solvent (tetrahydrofuran). The position of the peak, based on the calibration curve determined using polystyrene standards, does not correspond to the real MW, which is equal to 220 Dalton. Moreover, the UV spectrum shows that a considerable number of bitumen molecules have a higher elution time. This would mean that they are smaller than 220 Dalton, which is not realistic for the heaviest fraction derived from oil distillation. 

These considerations confirm the apparent character of the distributions, due to both the uncertainties in detection and the stationary phase not working as a pure molecular sieve. These effects are very important when dealing with bituminous materials, which, compared with polymers, vary greatly in the chemical composition of their constituents. Moreover, the level of molecular association is linked to operating parameters such as the flow rate, concentration, temperature and age of the solution. Therefore, these parameters, as well as the composition and porosity of the stationary phase, need to be carefully tuned before recording the results (see, for example, Branthaver et al. [18] and Yapp et al. [11]). In the case of bituminous materials, the choice is often that of preserving, at least partially, the molecular association. This is the case of a high-speed SEC, where both the flow rate and concentration of the solution are kept at higher values compared with the usual SEC [19,20], thus obtaining chromatographic peaks, which can be interpreted on the basis of different degrees of molecular association.

### 3.2. Inverse Mechanical Approach

As with GPC, the inverse mechanical approach also derives from the field of polymer science, where, from the very beginning, many efforts have been made to predict the viscoelastic properties from the knowledge of the MWD or average MW. In contrast, few studies have been carried out on the reverse problem, with the first seminal works dating to the late 1950s [7,8,21,22,23]. This inverse calculation of the MWD from the viscoelastic functions was also suggested as a useful way of obtaining MWDs for insoluble polymers, for which the traditional methods already available, such as light scattering, osmometry and GPC, were not applicable. The basic idea is that the viscoelastic response is strictly related to molecular motion, and thus depends on the complete spectrum of the relaxation times of the material. For this reason, several linear viscoelastic properties, such as the complex, storage or loss moduli, phase angle, relaxation spectrum or viscosity function, can, in theory, be deconvoluted to obtain the MWD.

The procedure suggested by Tuminello derives from the double-reptation mixing rule [24] and relates the phase angle (δ) to the MW:(10)$$\delta \left(\omega \right)=\int_{0}^{\infty }w\left(MW^{\prime }\right)c\left(MW^{\prime },\omega \right)dMW\prime$$
(11)$$c\left(MW\prime ,\omega \right)=\left[1-H\left(MW^{\prime }-MW\right)\right]$$
where *c(MW,ω)* is the monodisperse phase angle, *w(MW)* is the weight-distribution function, ω is the reduced angular frequency, and *H* is the Heaviside function. The last step is a correlation between the *MW* and *ω*, which, in the case of polymers, is well described by a simple mass law:(12)$$MW=k\omega^{-\alpha }$$
where *k* and α are constants. Introducing the new variable *x* = log *ω*, and differentiating Equation (12), the weight distribution as a function of the phase angle assumes the form:(13)$$w\left(M\right)=-\frac{10^{\alpha x}}{\alpha kln10}\frac{d\delta }{dx}$$

Likewise, a correlation between the *w(MW)* and other viscoelastic functions can be obtained. The first paper to apply the inverse method to bituminous binders was by Zanzotto and coworkers [9], who used a previously developed fractional form of the complex modulus (*G^*^*) [25]:(14)$$G^{*}\left(\omega \right)=i\eta_{0}\omega {\left[\frac{\prod_{k=1}^{m}\left(1+i\omega \mu_{k}\right)}{\prod_{k=1}^{n}\left(1+i\omega \lambda_{k}\right)}\right]}^{\beta }$$
which provides the following expression for the phase angle:(15)$$\delta \left(\omega \right)=\frac{\pi }{2}+\beta \left[\overset{m}{\underset{k=1}{\sum }}\arctan\left(\mu_{k}\left(\omega \right)\right)-\overset{m}{\underset{k=1}{\sum }}\arctan\left(\lambda_{k}\left(\omega \right)\right)\right]$$
where *i* is the imaginary unit, *β* is a constant *λ*, and *μ* are the relaxation times.

The details are reported by Zanzotto et al. [9], who also derived the values of *k* and α (Equation (12)) for bitumen by correlating the crossover frequency (*ω_c_*) at 0 °C with ${\bar{M}}_{n}$, obtained with osmometry for different bitumens. The correlation developed by Zanzotto et al. is given by Equation (16):(16)$$\log\left(MW\right)=2.880-0.06768\log\omega_{c}$$

The BMWD derived from this method clearly depends on the analytical form of the viscoelastic function. In the case of the fractional model, the δ−ω curve inevitably shows several inflection points that depend on the number of terms in Equation (14). Zanzotto et al. observed that, to obtain a representative description of the complex modulus and phase angle, the number of inflection points is quite limited, and some are directly observable in the experimental data. Nevertheless, this leads to a multimodal MWD that strongly depends on the number of terms used to fit the master curves.

A more recent approach assumes that the cumulative molecular weight distribution (CMWD) is proportional to the δ(ω) [10]. The hypothesis is that a direct correlation exists between the relaxation frequency (or time) of a molecule and its molecular weight. Below this frequency, the molecule does not contribute to the modulus, while, at a higher frequency, the opposite occurs. Therefore, the lower the frequency, the lower the number of molecules that contribute to the modulus, of which the dependence on the frequency strictly reflects the distribution of the molecular weights. An analogous consideration can be made for the phase angle that gradually moves toward liquid behavior, where all molecules are below their relaxation frequency. Since Zanzotto et al. already underlined that the phase angle is highly sensitive to the MW [9], the latter was chosen for the δ-method [10,13]. The details of the method are given in the original paper. However, the main assumptions are: (i) at a given frequency, the phase angle is directly proportional to the fraction of relaxed molecules; (ii) the cumulative molecular weight distribution is proportional to the value of the phase angle (when the phase angle reaches the value of 90°, the cumulative distribution is equal to the unity). Therefore, the BMWD can be obtained as the first derivative of the normalized master curve of the phase angle with respect to the log (MW):(17)$$w\left(MW\right)=\frac{d\left(\frac{\delta }{90}\right)}{d(\log\left(MW)\right)}$$

The authors adopted the frequency-molecular-weight correlation proposed by Zanzotto et al. (Equation (16)); thus, the master curve must be built with a reference temperature equal to 0 °C. Later, Cuciniello et al. proposed a different correlation, derived from GPC distributions instead of the osmometry average molecular weight [26].
(18)log(MW)=3.02−0.0870logω hard bitumen
(19)log(MW)=3.13−0.0951logω soft bitumen

Figure 3 shows a graphical representation of the method that formally coincides with the analysis of GPC data. The experimental data can either be the signal from the detector at the end of the GPC column or the first derivative of the phase angle with respect to the reduced angular frequency, which is then converted into the MWD by a calibration curve.

In both Equations (13) and (17), the shape of the distribution is calculated from the first derivative of the phase-angle master curve. Therefore, the final MWD depends on the mathematical procedure adopted for the differentiation. Considering the abovementioned difficulties, when using the fractional model (Equation (14)), the use of “smoother” rheological models has been suggested, such as the 1S2P1D [27] or 2S2P1D [28,29]. Of course, any model that accurately fits the master curve is possible, and each one will have specific features. It is also worth underlining that the choices of both the error function and minimization algorithm introduce a certain grade of subjectivity into the fitting procedure. However, the use of a model is not mandatory because the derivative of the normalized phase angle can be calculated numerically. In this case, the use of smoothing and noise-reduction algorithms can be helpful [30].

Additional considerations regarding the δ-method are necessary. The first derivative in Equations (13) and (17) implies that, if the phase-angle function does not monotonically decrease with the reduced angular frequency, then negative values of the *w*(*M*) will be obtained, which has no physical meaning. This can occur in the case of polymer-modified binders, which will be discussed later. Second, converting a phase-angle master curve into a cumulative molecular weight distribution is formally correct when the curve covers the whole range from 0 to 90°. 

While data in the terminal zone are easy to collect, the same is not true at high frequencies (low temperatures), and especially for bitumen, which becomes very brittle [31]. The available data must cover the widest possible range of temperatures; however, some are always missing. This results in a cut of the MWD, or else the rheological model can be used to extrapolate data outside the experimental range. Of course, extrapolation is another potential source of uncertainty in the apparent BMWD. For example, Figure 4 shows the experimental δ—master curve of a Pen 35/50 bitumen and the fit with the 2S2P1D model, which was subsequently used to calculate the BMWD reported in Figure 5. In both Figure 4 and Figure 5, it is possible to distinguish the data extrapolated outside the experimental range. All details regarding the material and experimental procedure to obtain the master curves are given in a previous publication [32].

## 4. Applications

GPC has been used in several studies on bituminous binders, and the recent review by Ma et al. provides a thorough overview of the available literature [1]. Here, we briefly analyze a few examples to show the differences and characteristics of the solution and bulk distributions.

### 4.1. Aging

Oxidative aging is one of the main causes of long-term pavement deterioration, and there is a vast amount of literature on this topic in the field of asphalt [34,35,36]. From the AMWD perspective, the difference between bulk and solution characterization is of paramount importance. Partial oxidation of bituminous molecules determines the formation of new functional groups, and, above all, carbonyl (C=O) and sulfoxides (S=O) [35]. These functional groups alter the polarity of the molecules, and thus their contribution to the cohesive energy of the material, as well as their degree of association with the surrounding molecules. In fact, oxidative aging is often characterized by a sort of shifting to a fraction with a higher polarity (for example, an increase in resins and asphaltenes with a decrease in the aromatic content with saturates, which tend to remain stable). This has a significant effect on the bulk properties that macroscopically show the hardening of the materials, with a consequent reduction in their strain tolerance. In contrast, the new functional groups have a modest effect on the molecular weight of the molecules. Therefore, the BMWD is expected to be more sensitive to aging than the SMWD. This is highlighted in Figure 6 and Figure 7, which show the effects of artificial aging on the SMWD and BMWD for the same unmodified base bitumen. 

In Figure 6, the SMWDs before and after aging are almost superimposable, and especially on the right-hand side, where they correspond to low-MW fractions. In contrast, aging stretches the BMWD toward high molecular weights, with a simultaneous variation in the relative intensity of the peaks (the meaning of which has been discussed in a previous work) [33]. 

Both distributions confirm that oxidation does not involve the less polar low-MW molecules, but mainly affects the aromatics, resins and asphaltenes. The asphaltenes undergo an increase in polarity and mutual interactions, which decrease their mobility and thus increase the apparent molecular weight. What is barely appreciable in solution becomes a well-visible effect in bulk. Consequently, several aging indexes have been proposed based on the evolution of the BMWD [13,32].

### 4.2. Additives or Modifiers

Many types of additives may be used to improve or restore the physical properties of bituminous binders, and, again, the SMWD and BMWD provide different information. Let us consider the case of rejuvenators, for example, which are vegetable oil added to an aged binder. Figure 8 shows the GPC spectra of a PAV-aged 50/70 penetration-grade binder, before and after the addition of a rejuvenating vegetable oil, as well as that of the oil.

The oil causes the appearance of a small shoulder (between approximately four and five minutes), as the interactions with the binder are hidden by the solvent. In this case, the GPC confirms, at least qualitatively, the presence of the additive, as the latter gives a signal at a specific elution time. In some cases (for example, isolated or well-visible peaks), a quantitative determination of the additive can also be obtained from the spectrum.

The BMWDs of the same two binders are given in Figure 9, together with that of the original unaged binder.

As expected, the oil affects the whole bulk distribution because it replaces the lost light fractions in order to restore the original degree of aggregation in the binder. Therefore, the main advantage of a bulk distribution is that it provides useful information. In Figure 9, a comparison between the original and rejuvenated binder shows that this goal is partially achieved, with the two distributions being similar, and overlapping in the high-MW region.

### 4.3. Polymer-Modified Bitumen (PMB)

The use of polymers to modify bitumen is well known. The polymer-modified bitumen (PMB) obtained is a blend of two constituents with very different average molecular weights, and, thus, in theory, they are easy to identify in the MWD. This is why GPC can assess the presence of polymers when there is a rapid elution peak that corresponds to the polymer, which is clearly distinguishable from the signal associated with the bitumen molecules [37,38]. 

The weight fraction of the polymer corresponds to the relative area of the peak in the spectrum, but this calculation can be quite difficult due to the small amount of loaded polymer. Moreover, due to both the presence of short chains in the polymer and interactions with the stationary phase, the polymer and bitumen signals cannot be completely separated. This can happen, for example, in the case of HS-SEC, which confirms that the appropriate operating conditions are necessary, depending on the aims of the analysis. A moderate flow rate and a highly diluted solution favor the separation of the peaks and are preferable when polymer content is required. The role of GPC in PMB is thus similar to the above-described example of low-molecular-weight additives. In Figure 10, the signal corresponding to the SBS copolymer is observable between 10 and 12 minutes of retention time.

Another potential cause of uncertainty in SBS-based PMBs derives from the common practice of adding sulfur to improve their storage stability and morphology [40]. Sulfur induces a chemical crosslinking of the polymer and the formation of covalent bonds between polymer and bitumen molecules [41]. This alters the length of the polymer chains and causes the inclusion of bitumen molecules in the signal attributed to the polymer. Moreover, the crosslinking may lead to insoluble polymer clusters that remain in the prefilter instead of being fed to the GPC column.

In contrast, the BMWD of a PMB provides useful information on the binder structure; however, the interpretation of the distribution may not be simple. If the polymer confers its elastomeric properties to the whole material, then it has a large effect on the relaxation time of the bitumen molecules. In the phase-angle master curves of PMBs, the elastic behavior induced by the polymer is manifested as a plateau in the intermediate viscoelastic zone. Before the plateau, at lower frequencies (higher temperatures), the material has a liquid-like behavior. At higher frequencies (lower temperatures), the rigid bituminous matrix tends to prevail in determining the overall mechanical properties, and the behavior shifts to solid-like. The extension of the time–temperature intervals in which these behaviors occur depends on the type of polymer, concentration and use of crosslinker. Because the plateau zone corresponds to low values of the first derivative of the phase angle in the frequency axis, it becomes the separation zone of a bimodal distribution in the molecular weight axis. The longer the plateau zone, the better the separation between the two modes of distribution. 

Figure 11a,b shows a master curve and the corresponding BMWD. The polymer is revealed by the high-molecular-weight peak, as in the GPC spectra; however, the relative area under this peak is much higher than the polymer content. This is because the polymeric network is swollen and includes bitumen molecules, the mobility of which is reduced, as in a gel [41]. The area of the peaks gives a quantification of the effect of the polymer on the overall distribution of the relaxation times and depends on the degree of the interactions between the polymer and bitumen.

The effect of the bitumen–polymer interactions on the relaxation times is clearly shown when comparing the BMWD distribution of a base bitumen and those of the corresponding PMBs obtained with and without the use of sulfur (Figure 12).

Without sulfur, the distribution of the PMB has the same range of molecular weights that correspond to the base bitumen, but the shape is very different. The presence of the polymer determines the abovementioned sharp separation between high- and low-molecular-weight zones, as the number of bituminous molecules involved in the network is proportional to the relative area of the second peak. When sulfur is added, the BMWD changes again: the second peak becomes larger and shifts significantly toward higher molecular weights, and the distributions show negative values between the low- and high-MW peaks. A larger area corresponds to a wider portion of the bitumen molecules involved in the polymeric network, which is the compatibilizing effect of sulfur. As a general observation, the more the peak is separated from the remaining part of the distribution, the higher the level of interactions and compatibility between the bitumen and polymer. The shift to the right is probably associated with the phase inversion, which is when the network assumes a sort of continuity and dominates the mechanical behavior. The whole material assumes an elastomeric character with higher relaxation times that reflect the behavior of the polymer. The negative peaks derive from the local minimum in the δ master curve. Although not acceptable from a mathematical perspective, the existence of negative weight fractions underlines a well-defined boundary between the prevalence of bitumen or polymer in determining the macroscopic behavior. Therefore, the negative values should be considered as important information on the material structure, instead of as a simple mathematical artefact.

Figure 13 shows the evolution of the BMWD in the PMB subjected to different levels of artificial aging. Again, there is a significant difference between the bulk and solution. The effect of aging on the distributions is different than that observed for the unmodified binder. There is almost no horizontal stretching, and the three peaks behave differently. The first one, which corresponds to a small MW, seems invariant; the second one, the intermediate MW, decreases; the third one, polymer-bond molecules, increases. Although the thermal degradation of the polymer causes chain breaking and weakens the network, the latter is still present, and the oxidative aging increases the number of molecules with reduced mobility.

## 5. Conclusions

The various analogies between polymers and bitumen enable similar characterization techniques to be used for both materials. However, if polymers have a homogeneous composition, then bitumen molecules cover a sort of continuous spectrum, from small nonpolar paraffinic chains to large and rigid asphaltenes with polar functional groups. Based on their interactions and aggregation, these molecules are arranged in very complex internal structures, which are often identified as colloidal and are determined by the relative abundance of the different components. The internal structure is, therefore, highly sensitive to the material composition and can change quite easily. In both polymers and bitumen, the structure and cohesive energy can be altered by parameters such as temperature, mechanical stress, oxidative aging and additives. However, in bitumens, there is an unstable colloidal equilibrium. This is why their behavior changes from Newtonian liquid to brittle solid within a fairly limited temperature range. The concept of the MWD of bitumen is thus different from that of polymers. The dimension of a single molecule contains less information than that of a polymer because its mobility also depends on the composition. Therefore, in relation to processing and in-service life, the molecules cannot be considered as single entities, but must be evaluated along with all their surroundings. The consequence is that, for bitumen, the difference between the bulk and solution MWDs is somewhat larger than it is for polymers. The bulk distribution has a closer relationship with the internal structure of the materials and is thus preferable when evaluating those properties that strictly depend on it, such as the degree of aging and the effect of additives/modifiers. The solvent separates the molecules; thus, the SMWD should be used when the size of the molecules is important (for example, in quantitative and qualitative evaluations of an added ingredient, such as a polymer or rejuvenator). Lastly, it is important to highlight that bulk and solution distributions clearly have many uncertainties, which means that, irrespective of the purpose, they should only be used to compare binders based on the same base bitumen.

## Figures and Tables

**Figure 1 materials-15-04700-f001:**
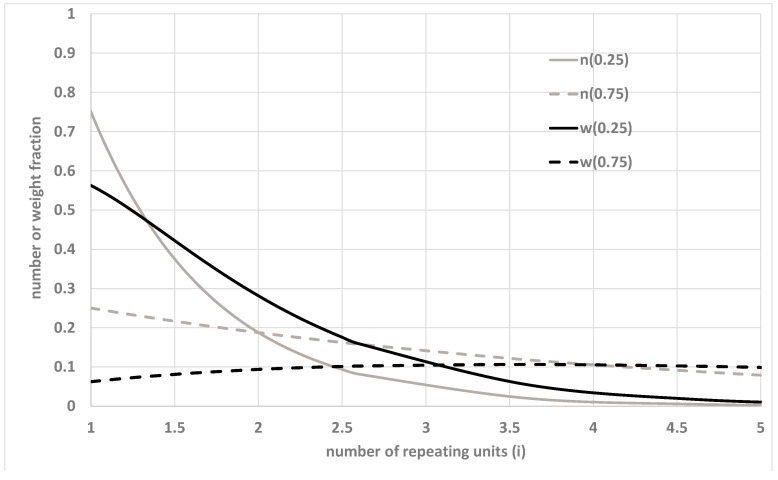
Number (n) and weight (w) distributions at a conversion equal to 0.25 and 0.75.

**Figure 2 materials-15-04700-f002:**
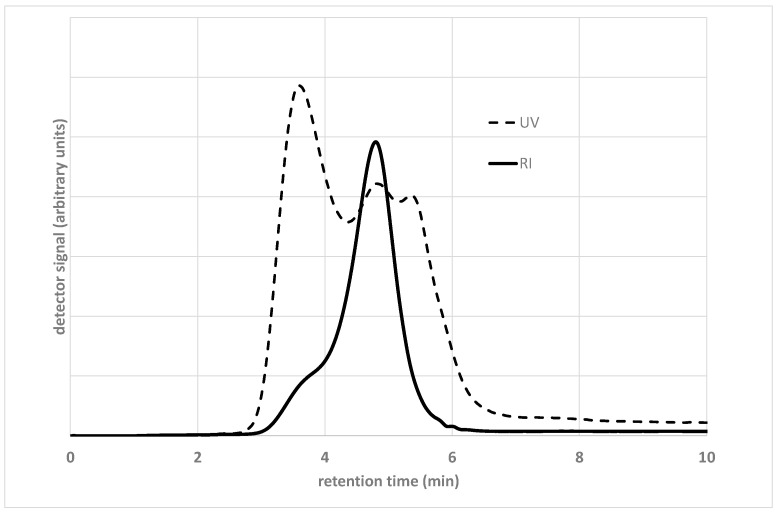
Distribution of retention time for the same sample of a base bitumen measured with UV and RI detectors. Details on the material are given in [16,17].

**Figure 3 materials-15-04700-f003:**
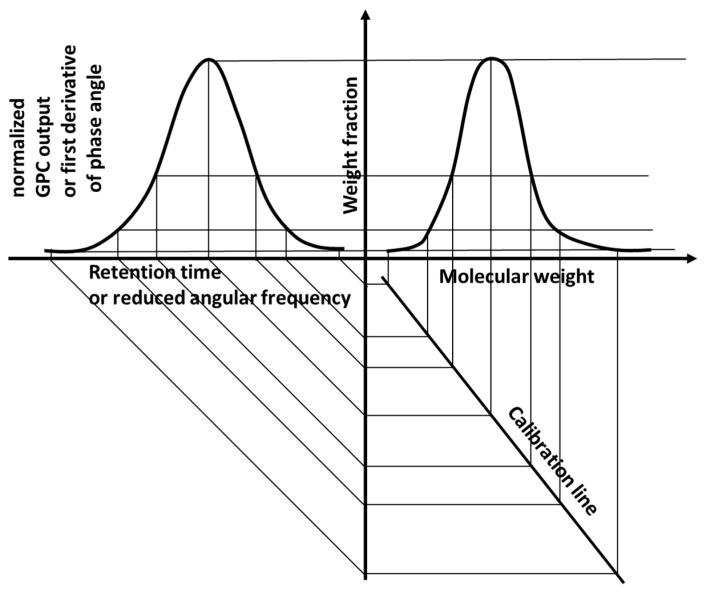
The conversion of the experimental data in an MWD.

**Figure 4 materials-15-04700-f004:**
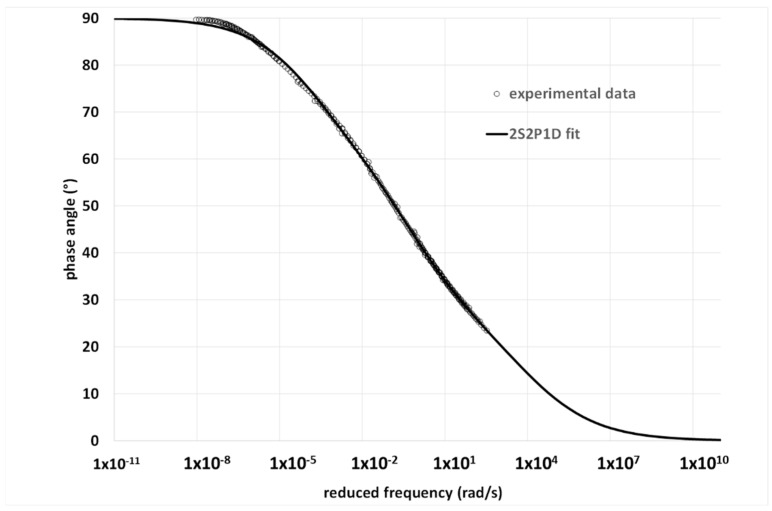
Phase-angle master curve of a Pen 35/50 bitumen. All details are given in [33].

**Figure 5 materials-15-04700-f005:**
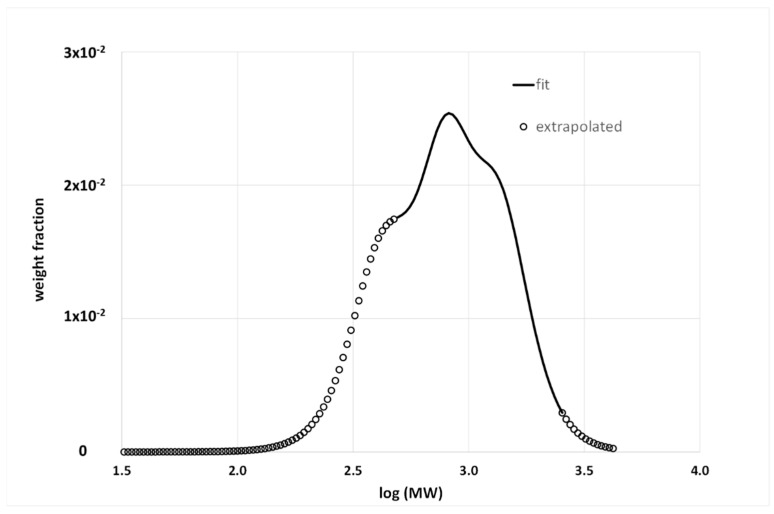
BMWD from the master curve reported in Figure 4.

**Figure 6 materials-15-04700-f006:**
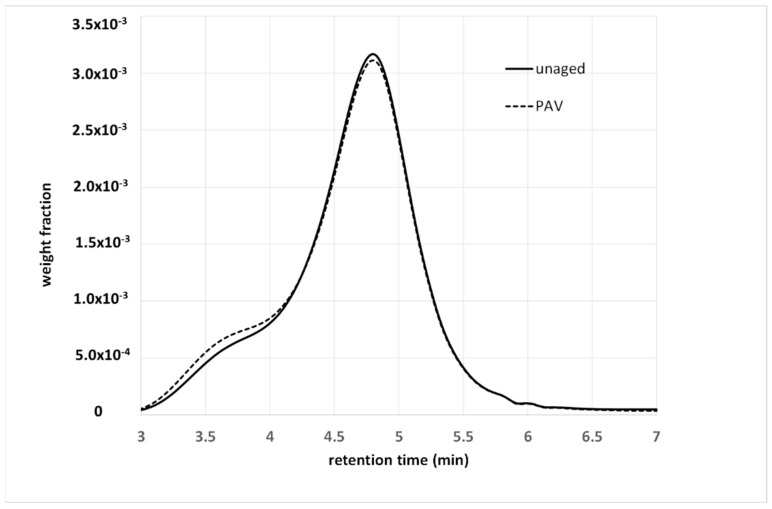
SMWD for a 35/50 Pen-grade base bitumen, before and after being subjected to a pressure-aging vessel (PAV).

**Figure 7 materials-15-04700-f007:**
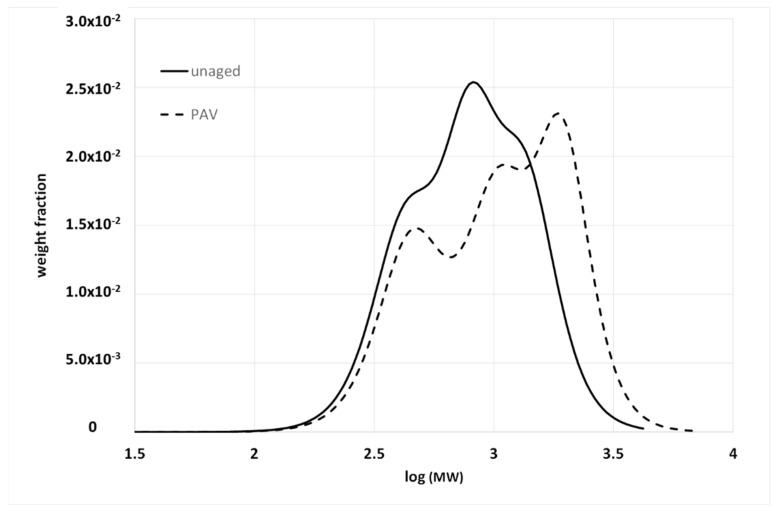
BMWD for a 35/50 Pen-grade base bitumen before and after PAV. All details are given in [33].

**Figure 8 materials-15-04700-f008:**
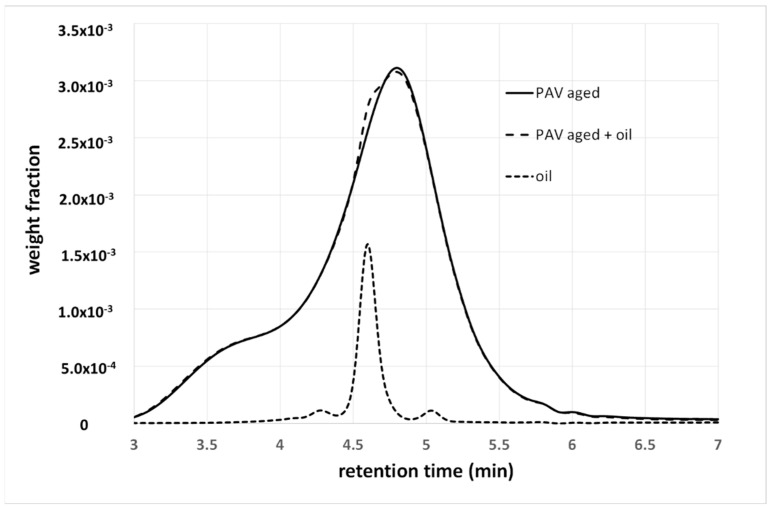
Weight fraction as a function of retention time of PAV-aged and rejuvenated binder, and of the rejuvenating oil. Details are given in [16,17].

**Figure 9 materials-15-04700-f009:**
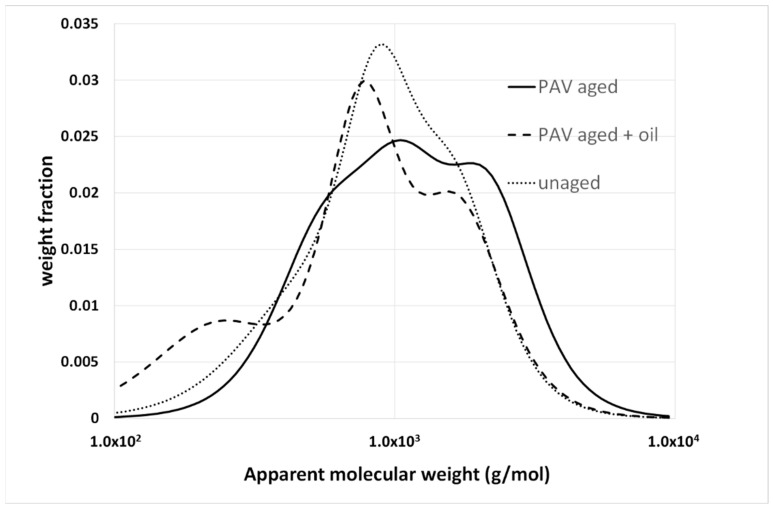
BMWDs of original PAV-aged and rejuvenated 50/70 binder.

**Figure 10 materials-15-04700-f010:**
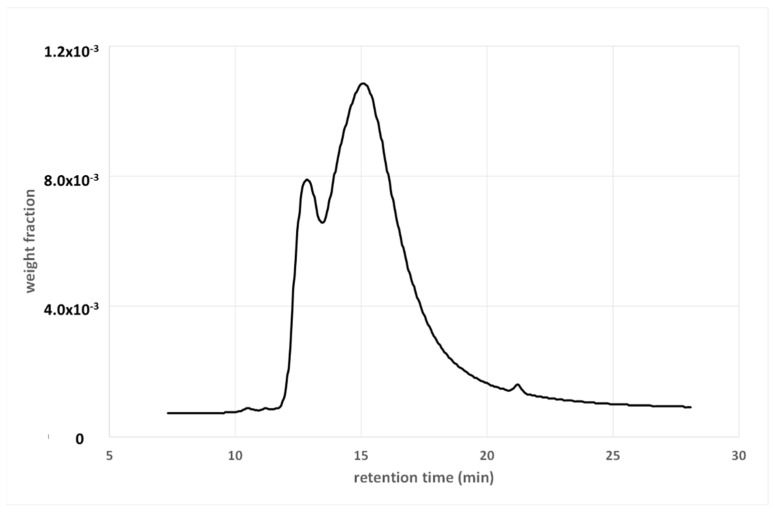
Weight fraction as a function of retention time of a PMB containing 6% SBS. Details on the materials and procedures are given in [39].

**Figure 11 materials-15-04700-f011:**
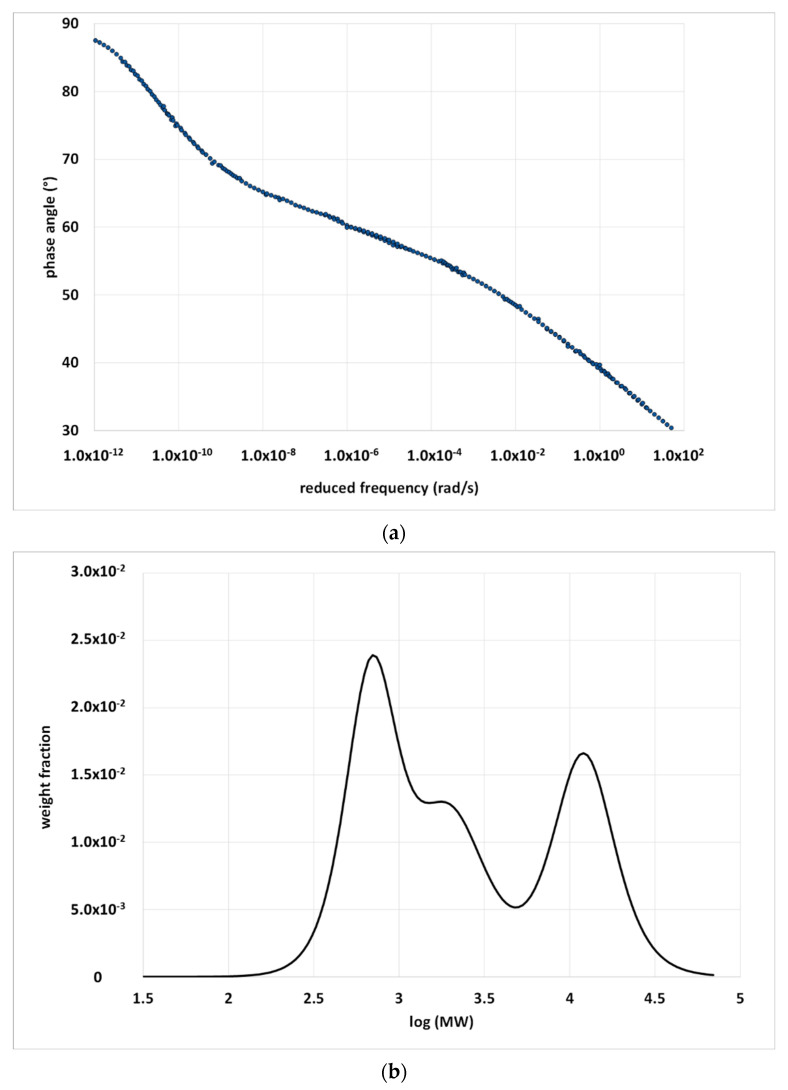
(**a**) Phase-angle master curve of a PMB containing 5% SBS. (**b**) BMWD obtained from the master curve reported in (**a**). Details are given in [32].

**Figure 12 materials-15-04700-f012:**
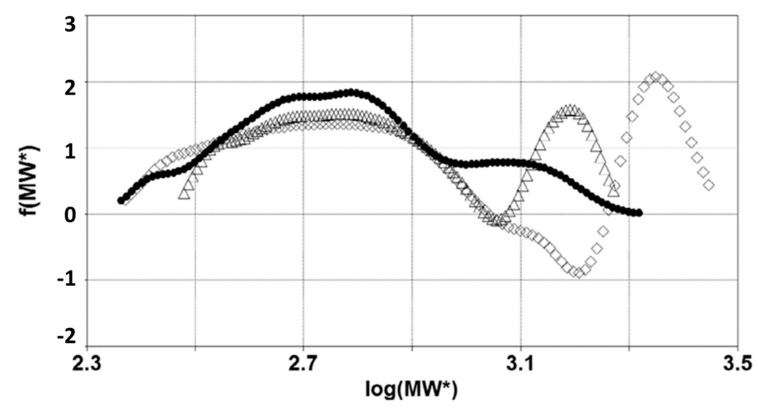
Apparent BMWD for a base bitumen before (●) and after modification with SBS (Δ), or SBS and Sulfur (◊). Image from [30].

**Figure 13 materials-15-04700-f013:**
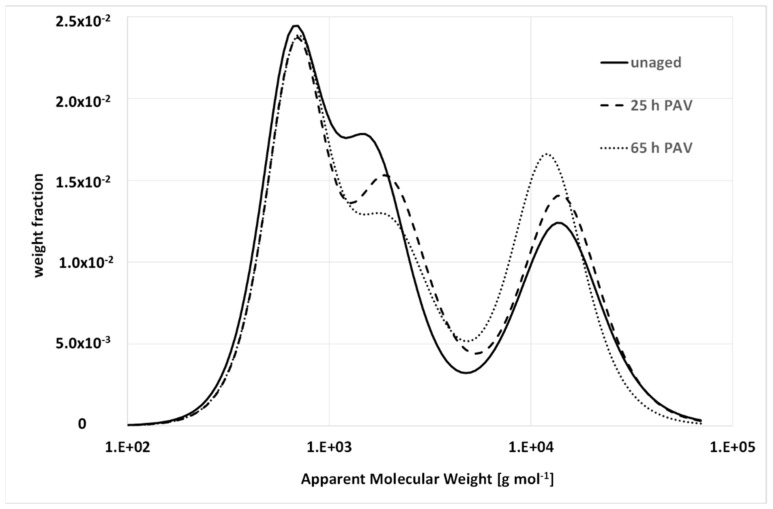
BMWD of a PMB with 5% SBS before and after prolonged artificial aging (25 and 65 hours of PAV, respectively). All details are given in [32].

## Data Availability

Original data are available upon request.

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
