# Peer review of "Apparent Molecular Weight Distributions in Bituminous Binders"

_materials, 2022, doi:10.3390/ma15134700_

Round 1

Reviewer 1 Report

In this review paper, the molecular weight distributions and their applications in evaluating the effects of aging or modifiers in bituminous binders are discussed. The paper provides comprehensive information on the pros and cons associated with different techniques to obtain the weight distributions. The authors discussed an interesting topic. The research findings are also valuable in guiding other researchers in the field to deploy proper techniques to investigate the effects of aging or modifiers in bituminous binders. However, there are margins for improvement. The reviewer requires an explanation or revision for the following comments:

1.      Both “Abstract” and “Conclusions” sections need to be revised/re-written to more clearly present the main findings of the research.

2.      The authors are recommended to add more references in the manuscript and discuss newly published articles to give readers a better insight on various approaches used to evaluate the effects of aging in bituminous binders, (e.g. https://doi.org/10.1177/03611981221088597, https://doi.org/10.1177/0361198120925807, and https://doi.org/10.3390/ma15124206 ).

3.      The figures font and format should be consistent throughout the manuscript.

4.      Page 16, line 461, replace “this the reason” with “this is the reason”.

5.      The authors are recommended to improve the readability of the materials presented throughout the manuscript either by proofreading or possibly rephrasing the long sentences to make sure the meaning is always getting across.

6.      Paper needs additional editing to correct the grammatical errors and punctuation mistakes.

Reviewer 2 Report

This is a complete work. I suggest publication.

Two important things should be cared:

1. some references from Prof. Formela highlight the importance of bitumen in compounding with polymers. Indeed polymers because of variable molecular weight should be cited. Some works of this professor should be cited, like:

https://doi.org/10.1016/j.conbuildmat.2016.12.134

2. english should be intensively modified. 

Reviewer 3 Report

A well written review, with a clearly identified scope (which is very important in those type of papers).

Some minor remarks.

1)      Math formalism and layout of equations, although understandable, could be more rigorous and precise. For example, differentials “d” shouldn’t be in “italics” and  “log” (L226)  is different from “ln” (Eq. 13), parentheses missing in Eq. 15.

2)      Most of Section 2 is derived from [4], there could be plagiarism risk

3)      Meaning of \beta in Eq 14 should be explained (fractional order of derivation?).

4)      In my pdf I get unreadable symbols on L223, 242

5)      L280, why “alternative”? 1S2P1D and 2S2P1D are also based on fractional derivation.

6)      L294, aren’t “brittle and fragile” synonyms?

7)      L296, when using rheological models another, I would say major source of uncertainty, is tha fact that fitting to experimental data is always subjective.

8)      About the “Applications” (especially aging and PMB): it would be interesting to remark if those data were obtained from binders recovered from asphalt mixture. Are there any literature data on that?

Finally, I highlight that the Authors present “Applications” only from their own works [8, 9, 27, 29, 30, 36, 38] which makes this section quite self-referential and thus less attractive.

Reviewer 4 Report

The paper is well written and easy to understand. Avoid questions as chapter titles, as mentioned in chapter 3 of this paper.

More references should be cited in the introduction, the authors have made many important observations, however, it would be good to cite some papers for the same.

Reviewer 5 Report

Thank you to the authors for submitting their work for consideration for publication. This is a nice work which examines the molecular weight distribution for bitumen based on mechanical tests, compared to the typical method used, GPC. The work is sound and very interesting. I just have two comments for the authors to consider:

1. How useful is the tool if it can only be used for a single base binder at a time?

2. The result seems to only be as good as the fit of the model? What about the issue of very complex phase angle master curves- for example, with bitumen which is modified with large amounts of crumb rubber? We do not have such good models to represent these materials yet.

Round 2

Reviewer 1 Report

The issues and comments raised earlier are addressed in the revised version of the manuscript.